# Neighborhood Social Environment and Body Mass Index: The Mediating Role of Mental Wellbeing

**DOI:** 10.3390/ijerph20166602

**Published:** 2023-08-18

**Authors:** Shayna D. Cunningham, Jennifer Mandelbaum, Fatma M. Shebl, Mark Abraham, Kathleen O’Connor Duffany

**Affiliations:** 1Department of Public Health Sciences, University of Connecticut School of Medicine, Farmington, CT 06032, USA; scunningham@uchc.edu; 2Department of Community Health, Tufts University, Medford, MA 02155, USA; jennifer.mandelbaum@tufts.edu; 3Medical Practice Evaluation Center, Massachusetts General Hospital, Boston, MA 02114, USA; fshebl@mgh.harvard.edu; 4Harvard Medical School, Boston, MA 02115, USA; 5DataHaven, New Haven, CT 06511, USA; mark@ctdatahaven.org; 6Department of Social and Behavioral Sciences, Yale School of Public Health, New Haven, CT 06510, USA

**Keywords:** neighborhood social environment, obesity risk, mental wellbeing, physical activity

## Abstract

The association between neighborhood-built environment and body mass index (BMI) is well-characterized, whereas fewer studies have explored the mechanisms underlying the relationship between neighborhood social environment and obesogenic behaviors. Using data from a random sample of 16,820 residents ≥18 years from all 169 Connecticut towns and seven ZIP Codes in New York, this study examines the influence of neighborhood social environment on residents’ mental wellbeing, physical activity, and BMI. Structural equation modeling was conducted to estimate direct and indirect effects of neighborhood social environment on BMI, using mental wellbeing and physical activity as intermediate variables. There were significant total [β(SE) = 0.741 (0.170), *p* < 0.0001], direct [β(SE) = 0.456 (0.1890), *p* = 0.016], and indirect [β(SE) = 0.285 (0.061), *p* < 0.0001] effects of neighborhood social environment on BMI. Low physical activity was a partial mediator of the effect of non-favorable neighborhood social environment on BMI [β(SE) = −0.071 (0.011), *p* < 0.0001]. The association between neighborhood social environment and BMI was also mediated by mental wellbeing [β(SE) = 0.214 (0.060), *p* < 0.0001], and by mental wellbeing through physical activity [β(SE) = 0.071 (0.011), *p* < 0.0001]. Study findings provide further support for building strong social environments to improve population health and suggest that strategies prioritizing mental wellbeing may benefit behavioral interventions aimed at reducing obesity risk and should be a focus of prevention efforts in and of itself.

## 1. Introduction

Numerous features of neighborhood-built environments (e.g., lack of green spaces, less access to grocery stores and supermarkets, low walkability indices) and social environments (e.g., lower levels of social capital and collective efficacy, higher levels of crime) are associated with increased likelihood of obesity [1,2,3,4,5,6]. While the effects of features of the built environment on body mass index (BMI) have been studied extensively, further research is needed on the mechanisms underlying the association between neighborhood social environment and obesogenic behaviors. Much of the scientific literature describing the neighborhood social environment and its association with BMI focuses on physical activity [7]. Less attention has been given to mental wellbeing, theorized to be a pathway influencing BMI both directly and indirectly via physical activity [7].

Multiple studies have demonstrated perceptions of neighborhood safety and social capital to be positively associated with physical activity among adults and children [8,9,10,11,12,13,14]. Neighborhoods with higher degrees of social disorder (which can include perceptions of cleanliness, trust, safety walking after dark, and vandalism) may have higher rates of obesity because individuals are less likely to engage in outdoor physical activities if they fear being attacked or robbed [15]. People who perceive their neighborhoods as unsafe have higher BMIs than those who think of their neighborhoods as safe [16]. The mechanisms underlying the association between social capital and weight status are less well understood and might include informal control and norms pertaining to health-related behaviors, collective efficacy, and social support [17]. Moreover, while findings from systematic reviews indicate that an association between neighborhood social capital and obesity exists, it depends on how social capital is defined and operationalized across studies and the covariates that are included [17]. For example, the potential effect of social capital on obesity may be influenced by other social determinants of health, such as socioeconomic status [17]. Individuals living in neighborhoods with greater inequality are more likely to have both higher energy intake and lower energy expenditure, putting them at risk for weight gain [18]. Social inequality has been found to be directly associated with obesity and to confound the association between the built environment and obesity [19].

Prior studies provide evidence that neighborhood social environment measures, such as social disorder, are associated with depressive symptoms [20,21]. A threatening neighborhood social environment may cause repeated physiologic responses that strain the body (otherwise called allostatic load) [15]. This chronic stress may inhibit weight regulation and healthy dietary patterns [22]. Living in a socially cohesive neighborhood has also been found to have a positive effect on mental health. For example, older adults living in neighborhoods with high collective efficacy were found to have lower prevalence of depression compared to those in low collective efficacy neighborhoods [23]. Similarly, high neighborhood social cohesion is associated with reduced depression [24,25]. The relationship between mental health status and obesity among adults is also well-documented [26,27]. A meta-analysis found that individuals with depression were 37% more likely to be obese compared to individuals without depression [27]. The association between depression and obesity may be explained through several mechanisms such as ineffective emotion regulation and stress response [28]. Chronic stress may lead to leptin resistance, reducing feelings of satiety [29]. Another way depression may lead to weight changes is through altered lifestyle habits [30]. For example, poor mental health and wellbeing (e.g., depression, stress, low mood) may present a barrier to physical activity [31,32].

Strengthening inferences about the effects of, and mechanisms through which, the neighborhood social environment influences BMI is essential for the development of effective place-based interventions. The present study utilizes a unique population-based sample to simultaneously examine the associations between neighborhood social environment, mental wellbeing, physical activity, and BMI. Mental wellbeing is conceptualized as a central mediator of the association between neighborhood social environment and BMI.

## 2. Materials and Methods

### 2.1. Data Source

This study is a secondary analysis of data from the 2015 DataHaven Community Wellbeing Survey [33]. The DataHaven Community Wellbeing Survey assesses quality of life and health among residents of Connecticut and adjacent sections of New York State, providing robust information on individual- and neighborhood-level wellbeing that was previously unavailable at local and state levels. Between April and October 2015, in-depth landline and cell phone interviews were completed with 16,820 randomly selected adults, aged ≥ 18, in all 169 towns in Connecticut and seven ZIP Codes in New York. Interviews were conducted by staff of the Siena College Research Institute in English or Spanish. Respondents’ residence in a qualifying town and ZIP Code was confirmed prior to survey implementation.

The landline sample, generated though random digit dialing, included both listed and unlisted numbers. Cell phone numbers were randomly selected from a list of dedicated wireless telephone exchanges from within Connecticut and the specified ZIP Codes within New York. Up to five contact attempts were made for each number selected. Residents were sampled on a proportional basis to match populations in large, midsize, and small cities and towns. Younger and lower-income towns were oversampled to include harder-to-reach populations. Data are weighted by age, gender, reported race, geographic area, and telephone status (i.e., landline, cell phone, or both) to ensure statistical representativeness.

### 2.2. Measures

Survey questions were drawn from previous local, national, and international studies, with significant input from local and national experts, e.g., [34,35,36,37]. These measures have been used by prior DataHaven Wellbeing surveys, as well as other surveys conducted in Connecticut [33,38]. Responses were reverse coded as needed, for consistent directionality, prior to being included as indicators in latent variables.

#### 2.2.1. Body Mass Index (BMI)

BMI was calculated based on self-reported weight and height, i.e., weight in kilograms divided by height in squared meters.

#### 2.2.2. Physical Activity

Respondents reported the number of days, in an average week, they exercised.

#### 2.2.3. Mental Wellbeing

Mental wellbeing was operationalized as a latent variable composed of three items. Participants indicated how often during the last month they were “bothered by feeling down, depressed or hopeless” using a Likert-type scale ranging from 1 (never) to 5 (very often). They were additionally asked, “Overall, how satisfied are you with your life nowadays,” and “how happy did you feel yesterday.” Responses ranged from 1 (not at all) to 5 (completely).

#### 2.2.4. Neighborhood Social Environment

Neighborhood social environment was operationalized as a second order factor composed of two sub-constructs: social capital and safety.

Social capital was composed of five items pertaining to trust in others and civic engagement. Respondents rated their level of agreement with the following statements: “People in this neighborhood can be trusted;” “Children and youth in my town generally have the positive role models they need around here;” and “If the fire station closest to your home was going to be closed down by your city or town, how likely is it that neighborhood residents would organize to try to do something to keep the fire station open?” Responses for the first two items ranged from 1 (strongly disagree) to 4 (strongly agree) and for the third, from 1 (very likely) to 4 (not at all likely). In addition, they responded ‘Yes’ or ‘No’ to the following question: “Over the past 12 months, have you volunteered for or through an organization or helped out as a volunteer to address needs in your community?” Lastly, they classified their “ability to influence local-government decision making,” on a scale ranging from 1 (great influence) to 4 (no influence at all). Higher scores indicate less social capital.

Neighborhood safety was composed of three items. Participants were classified as having been a recent victim of a crime if they answered yes to either of the following questions: In the past 12 months, “have you had anyone deliberately vandalize, try to steal, or steal any property that you own, or anyone attempt to break into your home” and “have you had an experience in which someone attacked you, tried to take something from you by force, or physically threatened you.” Additionally, respondents rated their level of agreement, ranging from 1 (strongly agree) to 4 (strongly disagree), with the following statement: “I do not feel safe to go on walks in my neighborhood at night.” Higher scores indicate a less safe environment.

#### 2.2.5. Neighborhood Built Environment

The following four items comprised the latent factor for built environment: (1) “Many stores, banks, markets or places to go are within easy walking distance of my home;” (2) “There are safe sidewalks and crosswalks on most of the streets in my neighborhood;” (3) “There are places to bicycle in or near my neighborhood that are safe from traffic, such as on the street or on special lanes, separate paths or trails;” (4) “My neighborhood has several free or low cost recreation facilities such as parks, playgrounds, public swimming pools, etc.”. Respondents rated their level of agreement with each statement, ranging from 1 (strongly agree) to 4 (strongly disagree).

#### 2.2.6. Control Variables

All analyses controlled for sociodemographic variables linked to perceptions of neighborhood social environment, mental wellbeing, and BMI in previous research, including self-reported age as continuous variable, and gender, race/ethnicity, and household income as categorical variables [1]. As evidence also suggests residence in neighborhoods characterized by high levels of deprivation may be negatively associated with mental health and BMI [21,39,40,41], a standardized neighborhood deprivation index (NDI) score was calculated for each respondent, based on residential census tract [42]. As described by Messer and colleagues (2006), the method entails using principal component analysis (PCA) on twenty census variables related to poverty, racial composition, household value, employment, occupation, education, housing, and crowding to obtain the first component [42]. Subsequently, variables with loadings > 0.25 were identified (12 variables), and only the variables with a lower confidence limit higher than the lower confidence limit of the median loading were retained (6 variables) and used in the final PCA model to create the NDI.

### 2.3. Data Analysis

Means with standard deviations (SD) and percentages were calculated to characterize the study population. Sample weights were provided for each record in the DataHaven survey dataset and were used in all analyses. In addition to demographic parameters, the Connecticut statewide sample was also weighted to match current patterns of telephone status (landline only, cell phone only, or both landline and cell phone), based on the state-level estimates from the 2013 National Health Interview Survey [43]. Post-stratification weighting was carried out separately for landline- and cell-phone-based surveys using 2013 US Census Bureau American Community Survey data [44], which were then merged, with appropriate calculation of final weights for each individual record.

Confirmatory factor analysis (CFA) was performed to examine the composition of the latent factors. An indicator reliability (i.e., the square of the correlation between the latent factor and its indicator) of >0.39 and a composite reliability (i.e., the internal consistency of indicators measuring a given factor) of >0.69 were considered as adequate values for inclusion [45].

where
*L_i_* = *standardized factor loading**Var*(*E_i_*) = *error variance associated with the indicator variable*

Structural equation modeling (SEM) was then employed to estimate direct and indirect effects of neighborhood social environment on BMI using mental health and physical activity as intermediate variables. Potential recursive relationships between physical activity and BMI and between physical activity and mental health were also assessed. Potential mediators and outcomes were regressed on the control variables (age, gender, race, income, NDI). Weighted least squares estimation with degrees of freedom adjusted for means and variances of latent and observed variables was used to determine how much of the extracted variance was captured by the factors. Model fit was assessed using significance of the factor loadings/path coefficients, residuals, and Comparative Fit Index (CFI), Tucker–Lewis index (TLI), and Root Mean Square Error of Approximating (RMSEA) [46,47,48]. Model fit indices were used as diagnostic tools to guide model improvement efforts, as suggested by Xia and Yang (2019) [49]. We elected to report unstandardized coefficients instead of standardized coefficients for multiple reasons, including (1) the indirect effects were calculated using the unstandardized coefficients, (2) to avoid confounding the unstandardized estimate with the sample variance, and (3) to enhance interpretability [50,51]. The CFA and SEM were performed using MPLUS version 7 [52]. All other analyses were conducted using SAS 9.4 [53].

## 3. Results

### 3.1. Descriptive Statistics

Response rates to the 2015 DataHaven Community Wellbeing Survey were consistent with industry norms for health and public opinion surveys. For example, 59% of the landline samples had unknown eligibility despite making more than five attempts to contact each number. Using the American Association of Public Opinion Response Rate calculation [35], which includes the sample with unknown eligibility in the denominator, the response rate was 9.0% for the landline sample and 5.7% for the cell phone sample [54]. Of the 16,820 live interviews, 4955 were completed on a cell phone and 11,865 on a landline.

Table 1 provides unweighted means, frequencies, and weighted percentages for survey participants sociodemographic characteristics. Respondents had a mean age of 57.63 years (SD = 17.57). Overall, 52% were female, approximately 3/4 (72.40%) were white, and 40.22% had an annual household income of $50,000 or less. NDI scores ranged from −1.79 to 4.19, with a mean of 0.17 (SD = 1.08).

Almost half (47.13%) of participants reported they have never been bothered by feeling down, depressed, or hopeless in the last month. Thirty percent (29.27%) were completely satisfied with their lives nowadays, and 39.12% were completely happy yesterday. Respondents exercised an average of 3.42 days per week (SD = 2.43); one-third (33.53%) exercised five or more days per week whereas 16.88% percent reported not engaging in physical activity. The average BMI was 27.78 (SD = 6.44).

### 3.2. Confirmatory Factor Analysis

Table 2 shows the results of the CFA, including standardized factor loadings and reliabilities. The CFA model had a moderate fit. Specifically, all the factor loadings were significant, the RMSEA was 0.045 [90% confidence interval = 0.044, 0.047], CFI was 0.782, and TLI was 0.731. Except for the four standardized factor loadings of 0.20, 0.20, 0.31, and 0.32, all loadings were 0.42 or greater. The highest was 0.90.

The indicators’ reliability varied from 0.04 to 0.80. The composite reliability ranged from 0.51 to 0.73, with the highest for the second-order factor of neighborhood social environment. The variance extracted by factors ranged from 0.26 to 0.75. For instance, the variance extracted for the neighborhood social environment factor was 0.75, indicating that 75% of the variance is captured by this factor and 25% is error.

### 3.3. Structural Model

The initial theoretical model had moderate goodness-of-fit indices: RMSEA was 0.042 (90%CI = 0.041, 0.434), CFI was 0.678, and TLI was 0.629. Several modifications have been made to improve model fit. More specifically, the recursive relationship between physical activity and BMI was assessed, and only the path from physical activity to BMI was retained. Similarly, the recursive relationship between physical activity and mental health was assessed, and only the path from mental health to physical activity was retained. In addition, NDI was added to the model as a predictor of the neighborhood social and the neighborhood-built constructs. Finally, paths that were not significant were removed; these included the path from physical activity to mental health and the path from age to physical activity. Model fit was assessed following each modification. For the final model, RMSEA was 0.033 (90%CI = 0.032, 0.034), CFI was 0.801, and TLI was 0.766.

Unstandardized final model results are shown in Figure 1. Neighborhood social environment (coded inversely) was significantly associated with mental wellbeing [β(SE) = −0.476 (0.028), *p* < 0.0001]. There was a significant total effect [β(SE) = 0.741 (0.170), *p* < 0.0001], direct effect [β(SE) = 0.456 (0.189), *p* = 0.016], and indirect effect [β(SE) = 0.285 (0.061), *p* < 0.0001] of neighborhood social environment on BMI. The association between neighborhood social environment and BMI was mediated by mental wellbeing [β(SE) = 0.214 (0.060), *p* < 0.0001], as well as by mental wellbeing through physical activity [β(SE) = 0.071 (0.011), *p* < 0.0001].

Neighborhood built environment (coded inversely) was significantly associated with neighborhood social environment [β(SE) = 0.534 (0.038), *p* < 0.0001). There were significant total and total indirect effect between neighborhood-built environment and BMI [β(SE) = 0.499 (0.094), *p* < 0.0001]. Five significant indirect pathways involving the neighborhood built environment were observed, three of which include neighborhood social environment (Figure 1): (1) neighborhood built environment to mental wellbeing to physical activity to BMI [β(SE) = 0.026 (0.006), *p* < 0.0001); (2) neighborhood built environment to mental wellbeing to BMI [β(SE) = 0.078 (0.027), *p* = 0.004); (3) neighborhood built environment to neighborhood social environment to mental wellbeing to BMI [β(SE) = 0.114 (0.033), *p* = 0.001); (4) neighborhood built environment to neighborhood social environment to BMI [β(SE) = 0.244 (0.102), *p* = 0.017), and (5) neighborhood built environment to neighborhood social environment to mental wellbeing to physical activity to BMI [β(SE) = 0.038 (0.006), *p* < 0.0001).

The model also included significant total, direct, and total indirect pathways between income and BMI [β(SE) = −0.346 (0.044), *p* < 0.0001, β(SE) = −0.231 (0.048), *p* < 0.0001, and β(SE) = −0.1155 (0.019), *p* < 0.0001, respectively). More specifically there were six indirect pathways, namely: (1) income to neighborhood social environment to mental wellbeing to physical activity to BMI [β(SE) = −0.006 (0.001), *p* < 0.0001); (2) income to neighborhood social environment to mental wellbeing to BMI [β(SE) = −0.017 (0.005), *p* < 0.0001); (3) income to mental wellbeing to physical activity to BMI [β(SE) = −0.017 (0.003), *p* < 0.0001); (4) income to neighborhood social environment to BMI [β(SE) = −0.037 (0.015), *p* = 0.016); (5) income to mental wellbeing to BMI [β(SE) = −0.052 (0.015), *p* < 0.0001); and (6) income to physical activity to BMI [β(SE) = 0.014 (0.006), *p* = 0.032).

## 4. Discussion

Findings from this study add to the evidence base demonstrating direct and indirect effects of neighborhood social environment on BMI. Moreover, they suggest mental wellbeing may be an important mediating pathway for this association, challenging the notion that prevention efforts should predominantly focus on lifestyle factors such as nutrition and physical activity. Rather, strategies prioritizing mental wellbeing may benefit behavioral interventions aimed at reducing risk of having a high BMI and should be a focus of prevention efforts in and of itself.

The association of mental wellbeing and BMI has largely been studied from a psychopathological perspective using measures for specific issues such as depression [55]. However, this approach does not acknowledge limitations of wellbeing that may not meet specific diagnostic criteria [55]. Measures of subjective wellbeing, like those employed by this study, are increasingly being used to inform and determine the success of policy decisions [56]. However, first it is necessary to identify factors that influence subjective wellbeing to determine where to best invest resources.

Consistent with previous research [56,57,58,59], social capital and safety represented key features of neighborhood social environments associated with mental wellbeing. These findings point to the benefits of neighborhood engagement and trust for both mental and physical health outcomes. Further, this study provides evidence that the neighborhood social environment may affect physical health outcomes (e.g., obesity) by improving mental wellbeing. It is possible that these associations are stronger at the hyperlocal level, as has been measured through spatial modeling of social capital, than they are at the level of the larger administrative boundaries such as census tracts that are most commonly used as the unit of analysis [60]. Other modifiable dimensions of the social environment warranting future research as intervention targets to promote mental wellbeing and, in turn, reduce individuals’ likelihood of being obese include social networks and norms, trust in public institutions, and racial segregation and discrimination [7].

This study has some limitations. The DataHaven Community Wellbeing Survey relies on self-report for most indicators. Self-report BMI in particular can be unreliable as respondents typically underestimate their body weight [61]. This may result in an underestimation of the proportion of study respondents classified as overweight or obese. Relatedly, the BMI itself is limited in fully predicting an individual’s obesity risk, though it has more utility as a tool for assessing population-level health as applied in this study [62]. Although consistent with similar telephone survey research [63], the low response rate may impact the generalizability of the findings if respondents and nonrespondents differ on the dimensions or variables of interest. The cross-sectional nature of the data precluded our ability to make causal inferences and to fully examine potential bidirectional relationships between mental health status, physical activity, and BMI. Information about the type and duration of physical activity was not available. Further, the primary aim of the DataHaven Community Wellbeing Survey did not relate to neighborhood social environments but rather to the plurality of environmental, psychosocial, and economic factors determining Connecticut residents’ health and wellbeing. We maximally used all relevant available data within the survey to construct the latent factors. However, our model still did not have the ideal fit. Therefore, future research should also account for other important features of social environments, such as social networks and norms, and risk factors for obesity, such as dietary behaviors [7]. Future studies should look to replicate our findings using similar Community Wellbeing datasets, in this community and nationally. Although we did not engage in open science practices for this study, such as pre-registration of the research questions and analysis plan, DataHaven provides open access to its data guides and de-identified datasets.

The strengths of this study include its large, population-based sample encompassing 16,820 adults across an entire state and the breadth of measures pertaining to neighborhood environment. Population-level estimates for neighborhood environments are not often available at this level. Using structural equation modeling, which simultaneously tests multiple potential mediators, enabled us to empirically examine the inherent complexity of associations between neighborhood social and built environments and BMI.

## 5. Conclusions

This study provides further support for building strong social environments that promote mental wellbeing to improve population health outcomes. With regard to obesity prevention, such interventions will require systems-level and multi-sectoral approaches and should be coupled with efforts to address the built environment [7]. Recognizing the diversity of potential causal pathways leading to higher likelihood of being obese, evaluations of these efforts should include their ability to promote mental wellbeing, in addition to increase physical activity, improve dietary behaviors, and a healthy BMI.

## Figures and Tables

**Figure 1 ijerph-20-06602-f001:**
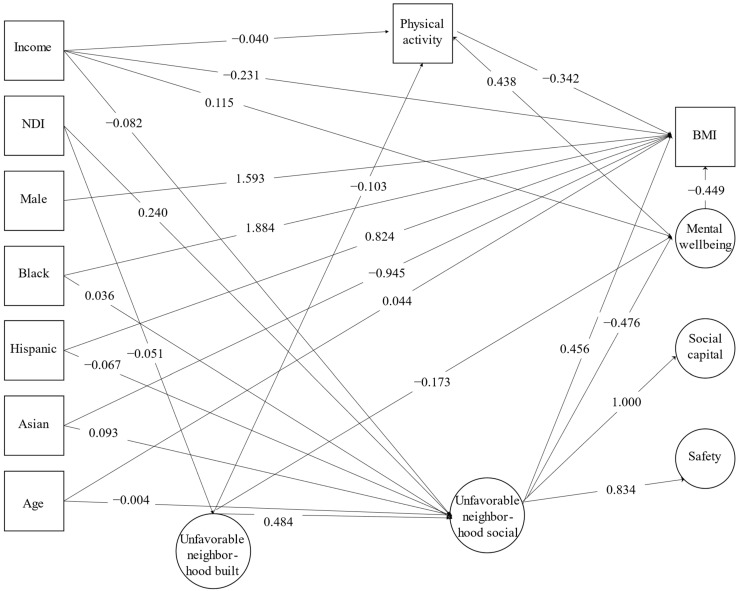
Unstandardized final model results for the associations between neighborhood social environment and residents’ mental wellbeing, physical activity, and BMI.

**Table 1 ijerph-20-06602-t001:** Sample sociodemographic characteristics (N = 16,219).

Characteristic	Unweighted Mean (SD) or Frequency	Weighted Percent
Age (years), (range = 18–94)	57.63 (SD = 17.57)	-
Gender		
Female	9053	52.10
Male	7166	47.90
Race		
White	11,822	72.40
Black	1698	8.70
Hispanic	1463	11.30
Asian	844	5.20
American Indian	392	2.40
Income		
Less than $15,000	1616	10.88
$15,000 to $30,000	1999	13.64
$30,000 to $50,000	2132	15.70
$50,000 to $75,000	2095	16.60
$75,000 to $100,000	1667	13.86
$100,000 to $200,000	2225	20.39
$200,000 or more	1052	8.92
Neighborhood Deprivation Index Score, (range = −1.79–4.19)	0.17 (SD = 1.08)	-

**Table 2 ijerph-20-06602-t002:** Reliability and validity of the constructs.

	Constructs and Indicators	Standardized Factor Loadings	Indicator Reliability	Composite Reliability	Variance Extracted
Mental wellbeing				0.73	0.48
	Overall, how satisfied are you with your life nowadays?	0.77	0.60		
	Overall, how happy did you feel yesterday?	0.69	0.47		
	During the last month, how often have you been bothered by feeling down, depressed, or hopeless?	0.61	0.37		
Neighborhood social environment (inversely coded: the higher the worse)				0.86	0.75
	Social capital	0.84	0.71		
	Safety	0.90	0.80		
Social capital				0.60	0.26
	Over the past 12 months, have you volunteered for or through an organization or helped out as a volunteer to address needs in your community?	0.20	0.04		
	How would you describe your ability to influence local-government decision making?	0.32	0.10		
	People in this neighborhood can be trusted.	0.65	0.43		
	Children and youth in my town generally have the positive role models they need around here.	0.75	0.56		
	If the fire station closest to your home was going to be closed down by your city or town, how likely is it that neighborhood residents would organize to try to do something to keep the fire station open?	0.43	0.18		
Safety				0.51	0.26
	In the past 12 months, have you had anyone deliberately vandalize, try to steal, or steal any property that you own, or anyone attempt to break into your home?	0.47	0.22		
	In the past 12 months, have you had an experience in which someone attacked you, tried to take something from you by force, or physically threatened you?	0.59	0.35		
	I do not feel safe to go on walks in my neighborhood at night.	0.47	0.22		
Neighborhood built environment (inversely coded: the higher the worse)				0.56	0.28
	Many stores, banks, markets or places to go are within easy walking distance of my home.	0.20	0.04		
	There are safe sidewalks and crosswalks on most of the streets in my neighborhood.	0.31	0.09		
	There are places to bicycle in or near my neighborhood that are safe from traffic, such as on the street or on special lanes, separate paths or trails.	0.69	0.48		
	My neighborhood has several free or low cost recreation facilities such as parks, playgrounds, public swimming pools, etc.	0.72	0.52		

## Data Availability

Data are available to qualified researchers upon reasonable request of the authors.

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
