# Peer review of "Neighborhood Social Environment and Body Mass Index: The Mediating Role of Mental Wellbeing"

_ijerph, 2023, doi:10.3390/ijerph20166602_

Round 1
Reviewer 1 Report
This study explores mediation role of mental wellbeing in the association between neighbourhood social environment and obesity risk among a sample of residents from all 169 Connecticut towns and seven ZIP codes in New York. The research is interesting and, overall, the study was well-conducted, but some questions and issues remained after a thorough reading of the manuscript.
Introduction
The background can be enhanced by providing more specific details about previous studies that have examined the association between neighborhood social environment (NSE), mental health, and obesity. It would be beneficial to include theoretical models and empirical evidence that explain the mechanisms underlying this association. Additionally, providing more context on NSE would be helpful. Furthermore, the knowledge gap could be clarified by acknowledging the existing evidence. You may want to consider removing the text on the bidirectional relationship between mental health and obesity as this cross-sectional study is aiming to explore the bidirectional relationship.
Methods
Measures.
1. BMI. Please specify whether it was treated as a continuous or categorical variable. Additionally, define the term "obesity risk" as it is used in the title and once in line 149 and a couple times in the discussion section. Given the diverse age and sex distribution of your sample, mention whether age- and sex-specific definitions for obesity were utilized.
2. Provide additional information about the type, duration, and other relevant details of the physical activity measures.
3. Mental wellbeing. If the mental wellbeing scale is a validated measure, please provide a reference. If not, describe the scale and clarify which aspects of wellbeing the five questions assess (e.g., depression, anxiety, overall mental health).
4. Figure 1 suggests the use of dummy variables for ethnicity. Please clarify this in the text.
5. For the neighborhood social environment variable, indicate whether a higher score indicates a better or worse environment. This clarification will help avoid confusion when interpreting the results.
Data analysis
6. Please describe the specific method used to weight the variables, as different approaches may yield different results.
7. Provide references for the cut-off values used to determine the reliability measures. It seems that a correlation of 0.624 (reliability of 0.39) and an internal consistency of 0.69 may not be optimal but still adequate. Specify the type of internal reliability measures used, such as Cronbach's alpha, to avoid ambiguity.
Results
8. It seems that the values for TLI and CFI do not fully support the fit of the model based on the cut-off 0.8 for these metrics as suggested in literature including the below paper. It seems that you have done some modification to improve the fit, but it is not clear what process. Although, the measures for the final model met the minimum.
Hu, L., & Bentler, P. M. (1999). Cutoff criteria for fit indexes in covariance structure analysis: Conventional criteria versus new alternatives. Structural Equation Modeling, 6, 1–55.
9. Clarify whether the item reliabilities mentioned in line 214 refer to the reliability if the item is removed from the scale. Provide the specific names or definitions of these measures in the methods section.
10. Elaborate on the modification process mentioned in line 224, providing details either in the methods section or in the results/tables following that line.
11. I wonder why authors reported unstandardized coefficients over standardized ones. Please justify for this.
12. The negative sign of the coefficient for NSE and mental wellbeing needs clarification, especially considering the labeling of latent variables in Figure 1. Determine whether a positive (negative * negative) social environment corresponds to a higher mental health score. To avoid confusion for readers, consider reversing the scoring of SN at the beginning. Just a suggestion.
13. The objective of the paper was examining the mediation role of well being in the association between NSE and obesity (or obesity risk), but in the results and Figure 1 we only see BMI and not obesity (risk). This needs to be clarified.
14. Figure 1 suggests that there are two indirect effects of NSE on BMI, one from mental wellbeing->BMI (obesity) and another one from SN->Mental well being->Physical activity ->BMI (obesity. Is this correct?
Discussion
15. Line 268, authors mentioned the association has largely been studies without citing any work. Please add some past research to here. It seems that psychological health has been used as an alternative for mental wellbeing. Please keep consistency of terminology used throughout the paper.
Conclusion
16. I would provide more specific conclusion according to the findings of the study. The current conclusion seems very general.
Minor comments
17. Add full caption for figure 1.
18. In line 49, what social disorders mean here. Since you give two example about depression you mean depression or mental health disorders?
19. Line 211. There is a extra number (36 ) in the CI bracket.
Author Response
thank you for your suggestions, please see attachment

Reviewer 2 Report
This is an interesting study that draws on a valuable large-scale dataset. I have a number of comments below that the authors should consider.
1. At the beginning of the introduction section, it would be helpful to give a more nuanced discussion of this literature acknowledging mixed/ inconsistent findings– a few examples: for the built environment https://ij-healthgeographics.biomedcentral.com/articles/10.1186/s12942-021-00260-6 , for the local food environment - https://onlinelibrary.wiley.com/doi/epdf/10.1002/oby.21118 and for social environment - https://onlinelibrary.wiley.com/doi/full/10.1111/obr.12760
2. There is a need for more specificity around where measures have come from as well as their justification for why they are suitable in the current context.
3. I was surprised that the CFI and TLI were a little lower than is often deemed ideal. Ideally, can the model be further refined to enhance the CFI and TLI? If not, the authors should explicitly interpret and justify why these are acceptable in the current context.
4. In the discussion section, more specific discussion of the shortcomings of self-reported BMI (e.g., https://bmcobes.biomedcentral.com/articles/10.1186/s40608-016-0102-8) and then BMI itself as a measure of obesity is warranted given this is a key outcome measure.
5. Finally, was any kind of open science practice engaged in? For example, pre-registration of the research questions and analysis plan. This is ideal for secondary data analysis of this kind. If not, this should be mentioned as a limitation and should be considered by the authors for future studies.
Author Response

(The authors gave the same response as above.)

Round 2
Reviewer 1 Report
I find that the authors have responded to my critiques with careful and comprehensive explanations. I value the supplementary context and the thorough attention given to my feedback, particularly more detailed information on variables and methodologies employed.
My only comment has to do with getting clear on the cut-off value for internal consistency from the following comment:
Provide references for the cut-off values used to determine the reliability measures. It seems that a correlation of 0.624 (reliability of 0.39) and an internal consistency of 0.69 may not be optimal but still adequate. Specify the type of internal reliability measures used, such as Cronbach's alpha, to avoid ambiguity.
Authors provided references for the internal consistency cut-off. I acknowledge absence of consensus regarding cut-offs for internal consistency measures, however, I perceive that none of the suggestions favor a value lower than 0.7 as the preferable range. The authors could consider a term that carries a somewhat less intense connotation than "optimum."
Author Response
Thank you for your advice, please see attachment.
